# The Effect of a Planetary Health Diet on the Human Gut Microbiome: A Descriptive Analysis

**DOI:** 10.3390/nu15081924

**Published:** 2023-04-16

**Authors:** Jacqueline Rehner, Georges P. Schmartz, Tabea Kramer, Verena Keller, Andreas Keller, Sören L. Becker

**Affiliations:** 1Institute of Medical Microbiology and Hygiene, Saarland University, 66421 Homburg, Germany; jacqueline.rehner@uks.eu (J.R.);; 2Chair for Clinical Bioinformatics, Saarland University, 66123 Saarbrücken, Germany; 3Department of Medicine II, Saarland University Medical Center, 66421 Homburg, Germany

**Keywords:** microbiome, Planetary Health, metagenomics, diet, dietary fiber

## Abstract

In 2019, researchers from the EAT-*Lancet* Commission developed the ‘Planetary Health (PH) diet’. Specifically, they provided recommendations pertaining to healthy diets derived from sustainable food systems. Thus far, it has not been analysed how such a diet affects the human intestinal microbiome, which is important for health and disease development. Here, we present longitudinal genome-wide metagenomic sequencing and mass spectrometry data on the gut microbiome of healthy volunteers adhering to the PH diet, as opposed to vegetarian or vegan (VV) and omnivorous (OV) diets. We obtained basic epidemiological information from 41 healthy volunteers and collected stool samples at inclusion and after 2, 4, and 12 weeks. Individuals opting to follow the PH diet received detailed instructions and recipes, whereas individuals in the control groups followed their habitual dietary pattern. Whole-genome DNA was extracted from stool specimens and subjected to shotgun metagenomic sequencing (~3 GB per patient). Conventional bacterial stool cultures were performed in parallel and bacterial species were identified with matrix-assisted laser desorption/ionization time-of-flight (MALDI-TOF) mass spectrometry. We analysed samples from 16 PH, 16 OV, and 9 VV diet patterns. The α-diversity remained relatively stable for all dietary groups. In the PH group, we observed a constant increase from 3.79% at inclusion to 4.9% after 12 weeks in relative abundance of *Bifidobacterium adolescentis*. Differential PH abundance analysis highlighted a non-significant increase in possible probiotics such as *Paraprevotella xylaniphila* and *Bacteroides clarus*. The highest abundance of these bacteria was observed in the VV group. Dietary modifications are associated with rapid alterations to the human gut microbiome, and the PH diet led to a slight increase in probiotic-associated bacteria at ≥4 weeks. Additional research is required to confirm these findings.

## 1. Introduction

In 2019, the EAT-*Lancet* Commission developed the so-called ‘Planetary Health diet’ (PH), a diet concept framework, which could provide a healthy diet for up to 10 billion people in 2050 within the planetary boundaries from sustainably sourced food, thereby reducing the worldwide number of deaths associated with a poor diet. The main focus of this diet consists of a reduction in animal products and processed food consumption and an increase in dietary fibre uptake through plant-based products [1,2]. 

Dietary fibre is a non-digestible carbohydrate for humans, but a main nutrient source for bacteria, which reside in the human intestine. The human gut microbiome describes all such microorganisms and their genomic information, including bacteria, viruses, fungi, and archaea, which are located in several niches in the gastrointestinal tract [3]. Its impact on health homeostasis and risk-modulating role in developing a variety of chronic, especially inflammatory, diseases, as well as disease progression, have been evaluated in a recent study [4]. The bacterial composition within the human gut can be altered, especially in the first three years of life, but also later on during adulthood. Major microbiome-influencing factors include the mode of delivery (i.e., natural passage through the birth canal or caesarean section), early life nutrition, as well as stress and diet choices during adulthood [5,6]. Focusing on diet and its consequences for bacteria-derived metabolites produced in the gastrointestinal tract, dietary fibre has been shown to be one of the main modulating nutrients [7]. 

Commensal members of the gut microbiota ferment these poly- and oligosaccharides, thereby producing short-chain fatty acids (SCFAs) such as acetate, propionate, and butyrate. SCFAs have been shown to influence glucose and lipid metabolism and regulate immunity, inflammation, and blood pressure [8]. Furthermore, the presence of SCFA-producing bacteria, and thus also the presence of SCFA detected in faeces, has been correlated with a protection against allergic reactions in the respiratory tract, suggesting their important role in shaping the immune system [9]. Hence, SCFAs are key elements in health homeostasis [10,11,12]. Dietary fibre uptake has further been correlated with a greater gut microbiota diversity and, if compared with the Western diet, less occurrence of chronic inflammatory disease through SCFA-producing gut microbiota [13,14]. Therefore, an increase in dietary fibre intake, as suggested by the PH diet, can lead to an increase in microbial-derived SCFAs, which have a positive and protective effect on overall health. 

The Mediterranean Diet, which focusses on an increase in dietary fibre uptake through plant-based foods and, similar to the PH diet, a reduction in processed foods and saturated fatty acids, moderate consumption of fish, poultry, and dairy products, as well as low consumption of red meat, was shown to positively influence the human gut microbiome and overall health status. The consumption of animal-derived foods is clearly reduced in the Mediterranean Diet when compared with the Western diet. However, the PH diet concept suggests to reduce the intake of meat and dairy even further. After following the Mediterranean Diet, an increase in microbiota diversity and microbiota-derived metabolites, in particular SCFAs, has previously been reported [15]. 

Another diet concept that is gaining more popularity is the plant-based diet. This diet emphasizes the consumption of plant-derived foods, such as fruits, whole grains, nuts, seeds, and vegetables, whereas animal products are minimised or strictly eliminated [16]. Similar to the Mediterranean Diet, following a plant-based diet has been shown to increase microbial diversity in the human intestine and positively affect the abundance of beneficial bacteria, such as *Prevotella* sp. [17]. Moreover, plant-based diets have been associated with reduced inflammation, lower risk of cardiovascular diseases, and improved glucose metabolism [18]. 

Another popular approach to maintain overall health, as well as weight management, is a low-fat diet. These diets usually focus on reducing the intake of fat to a maximum of 30% of total energy intake, while on the other hand increasing the consumption of other macronutrients, such as protein, carbohydrates, and dietary fibre [19]. Low-fat diets can be a powerful method in weight management; however, they also have been shown to decrease the diversity and abundance of several beneficial bacteria in the gut, such as *Bifidobacterium* sp. As these diet concepts vary greatly in the specific composition of the chosen foods and nutrients, positive changes within the intestinal microbiota composition have also been reported, such as an increase in beneficial *Prevotella* sp., similar to the results after following a Mediterranean Diet [20,21].

The focus of the PH diet consists of an increase in dietary fibre through the consumption of vegetables, fruits, and whole grains, and could thus lead to similar changes within the human gut microbiome as, for example, the Mediterranean Diet or plant-based diet concepts. While the PH diet concept is gaining more and more attention and support from various stakeholders, e.g., pertaining to an improved cognitive function, criticism has been raised about a relative lack of scientific evidence pertaining to its actual health effects [22,23,24]. To shed light on the controversial discussion about the PH diet concept, we aimed to analyse the effects of following the PH diet over the course of twelve weeks on overall biodiversity and gut microbiota composition in contrast to the most prevalent omnivorous diet (OV) and the vegan/vegetarian diet (VV). The OV Western diet followed by the participants consisted of a low intake of dietary fibre through fruits, vegetables, and wholegrains. Furthermore, individuals following this diet concept had a very high intake of highly processed foods, dairy products, meat, and refined sugars, forming the opposite of the PH diet concept. Individuals following a vegan diet are characterised by the eradication of any animal-derived products as nutrient sources; however, levels of dietary fibre intake and highly processed foods vary greatly between individuals. The abdication of meat products from an individual’s diet concept is the central component of the vegetarian diet, which was included in the VV as well. Yet, similar to individuals following a vegan diet, ranges of dietary fibre uptake and highly processed foods can vary.

## 2. Materials and Methods

### 2.1. Study Design

Healthy adults aged ≥ 18 years were recruited to the study. Volunteers were invited to participate in the Saarland area, southwest Germany from January to April 2022. Several exclusion criteria were defined to reduce potential bias owing to the relatively small number of study participants, i.e., pregnancy, active smoking, acute and/or chronic disease conditions, and the use of antibiotics within the last 6 months prior to inclusion. We recorded a detailed medical history of each participant, including major factors that affect the microbiome, such as (i) birth condition, (ii) medication during the first three years of life, (iii) exposure to animals within the first three years of life, and (iv) breast milk or formula use. Participants were divided into three groups according to their diet: two control groups, following a VV or OV for at least one year, and the intervention group. Participants belonging to the intervention group changed from an omnivorous diet to the PH diet. Prior to the study, these participants received detailed instructions and recipes according to the guidelines developed by the EAT-*Lancet* commission (document available online at https://www.wwf.de/fileadmin/fm-wwf/Publikationen-PDF/Landwirtschaft/wwf-wochenmenue-besseresser-innen-flexitarisch.pdf, (accessed on 12 April 2023). All participants collected faecal samples in a sterile collection tube at four different time points: initiation of the study and after two, four, and twelve weeks (Figure 1). Samples were then transferred to the laboratory within 24 h and stored at −80 °C until further processing. Furthermore, we asked all participants to document whether they had an excessive alcohol intake during the course of the study, as well as the exact foods they consumed two days prior to the collection of each faecal sample in a printed food diary, in order to reduce any potential bias that might be explained by different food choices shortly before sample collection. Individuals adhering to the PH diet were asked to track any divergence from the foods recommended by the EAT-*Lancet* commission across the entire study duration.

### 2.2. Ethical Considerations

All faecal samples were collected at the Saarland University Medical Center (Homburg, Germany) after having obtained written informed consent from all participants. For this study, we obtained ethical approval from the regional ethics committee (‘Ärztekammer des Saarlandes’, reference no.: 116/22).

### 2.3. DNA Extraction

We extracted whole-genome DNA from all faecal samples using the ZymoBIOMICS DNA Miniprep Kit [25]. DNA was isolated and purified according to the manufacturer’s protocol. Briefly, 50 mg of faecal matter was used for the mechanical lysis step of the protocol, according to the manufacturer’s recommendation. The respective lysis of microbial cells was performed using the MP Biomedicals™ FastPrep-24™ 5G Instrument (FisherScientific GmbH, Schwerte, Germany). The manufacturer’s protocol was adjusted in regards to the used velocity and duration of the mechanical lysis, which was increased to 6 m/s for 45 s three times with 30 s of storage on ice in between each lysis step. Finally, we eluted the DNA in 20 µL of DNase-/RNase-free water. Subsequent concentration determination of the eluted DNA was performed via NanoDrop 2000/2000c (ThermoFisher Scientific, Wilmington, NC, USA) full-spectrum microvolume UV/Vis measurements.

### 2.4. Library Preparation and Sequencing

Extracted whole-genome DNA was sent to Novogene Company Limited (Cambridge, UK) for library preparation and sequencing. Briefly, samples were subjected to metagenomic library preparation and further sequenced via paired-end Illumina Sequencing PE150 (HiSeq). For all samples, 3 Gb reads per sample were generated.

### 2.5. Culturing of Bacteria

Native samples from five randomly selected participants per diet group were homogenised by vortexing after defrosting in order to achieve equal bacterial distribution within the sample without lysing the cells. Then, samples were streaked out on three different agar plates: tryptic soy agar with 5% sheep blood (TSA), MacConkey (MC), and Columbia (Co) agar plates (Becton, Dickinson and Company, Franklin Lakes, NJ, USA). We incubated all TSA and MC agar plates at 35 °C and 5% CO_2_ for 18 h to 24 h. Anaerobic bacteria were cultivated on Co agar plates in an anaerobic environment at 35 °C for at least 48 h.

### 2.6. Mass-Spectrometry-Based Identification

After incubation of native sample material on different agar plates, grown bacterial colonies were identified on the species-level using matrix-assisted laser desorption/ionization time-of-flight (MALDI-TOF) mass spectrometry (MS). To this end, we picked colonies and spotted them onto the MALDI-TOF target plate and overlaid them with 1 µL of α-cyano-4-hydroxycinnamic acid (CHCA) matrix solution (Bruker Daltonics), which is composed of saturated CHCA dissolved in 50% (*v*/*v*) of acetonitrile, 47.5% (*v*/*v*) of LC-MS grade water, and 2.5% (*v*/*v*) of trifluoroacetic acid. The overlaid spots were then dried at room temperature and the target was subsequently placed into the Microflex LT Mass Spectrometer (Bruker Daltonics, Billerica, MA, USA) for MALDI-TOF MS analysis. We performed all measurements with the AutoXecute algorithm using FlexControl^©^ software (version 3.4; Bruker Daltonics, Billerica, MA, USA). Each spot was automatically excited with 240 laser shots at six random positions to generate protein mass profiles in linear positive ion mode. The laser frequency was set to 60 Hz, high voltage of 20 kV, and a pulsed ion extraction of 180 ns. We measured mass charge ratio ranges (*m*/*z*) from 2 kDa to 20 kDa. The MALDI BioTyper software was used to identify bacterial species based on their protein mass profiles measured. In this study, we only considered identification scores ≥ 2.0 for analyses, which represent a precise identification on the species level, while scores between 1.7 and 1.99 were discarded as they are considered as possible species identification, and all identification scores below 1.7 were considered unsuccessful identification.

### 2.7. Data Analysis

The first step of data analysis comprised human read removal with KneadData (version (v):0.7.4, command line arguments (cla): “--trimmomatic-options=’LEADING:3 TRAILING:3 MINLEN:50′ --bowtie2-options=’--very-sensitive --no-discordant –reorder’”) [26]. Next, we visualised the quality of the remaining reads with fastp (v:0.20.1) and MultiQC (v1.11) on default settings [27,28]. We computed a first taxonomic profile of quality-controlled reads with MetaPhlAn3 (v3.0.13, cla: “-t rel_ab_w_read_stats --unknown_estimation --add_viruses”) on the ChocoPhlAn (v:mpa_v30_CHOCOPhlAn_201901) resource [29]. A second taxonomic profile was generated based on sourmash (v4.4.3, cla: “sketch dna -p k = 21, k = 31, k = 51, scaled = 1000, abund --merge”) and the prepared Genome Taxonomy Database (v:GTDB R07-RS207 all genomes k51) [30,31]. Sample signatures were computed for k-mer sizes 21, 31, and 51. Distances among samples and database comparison were computed using k-mer signatures of size 31 and 51, respectively. All taxonomic profiles were then pruned and rescaled to remove viral counts.

The results of the individual samples were aggregated, and further downstream analysis was performed in R relying on the phyloseq package (v1.40.0) [31]. β-diversity was computed using the weighted UniFrac distance. Shannon diversity was used as the α-diversity measure and a two-sided unpaired Wilcoxon rank sum test was performed to test significance with a false discovery rate of 0.05. The two-dimensional embedding of sourmash sketches was performed with UMAP (v:0.2.8) [32].

Differential abundance analysis was performed with ALDEex2 (v:1.28.1) and ANCOMBC (v:1.6.2) comparing vegetarians and omnivores [33,34]. MetaPhlAn3 relative taxonomic abundances were scaled by their read count of the sample after quality control for ANCOMBC. A mean species abundance across all time points was computed for each participant, adjusting for library size if absolute counts were considered. Further, for a species to be considered for analysis, it had to be detected in over 10% of samples. Next, abundance analysis was performed, and the results were sorted by absolute effect size. We pruned the list, focusing only on the first ten percent, and intersected the sets derived from the same taxonomic profiles.

## 3. Results

### 3.1. Intestinal Microbial Diversity Stays Relatively Stable over Time

Overall, 41 individuals from the same geographic location (Germany) were included: 16 participants following an OV, 9 following a VV, and 16 individuals who changed from an OV pattern to the PH diet at inclusion. Participating individuals were between 19 and 59 years old, with age ranges between all diet groups being non-significantly different (ANOVA *p*-value ≈ 0.84). Sex ratios differed significantly between the three diet groups (Fisher’s exact test *p*-value ≈ 0.024), with more females in the VV group (8/9 individuals). General information about age, sex, and body mass index (BMI) is summarised in Table 1.

Over the course of twelve weeks, the average α-diversity remained relatively stable for all diet groups (Figure 2A). Slight increases and decreases for individual participants were detectable between the different time points. On the one hand, investigation of the β-diversity based on dimensionality reduction of species information showed no distinct cluster formation, suggesting that, independent of the diet and time point, samples were all rather similar in their microbial composition (Figure 2B). On the other hand, reference-free diversity analysis based on sequence information alone with sourmash highlighted VV samples to be similar, whereas samples from OV and PH did not form distinct clusters, suggesting similarities between those two groups (Figure 2C) [29].

### 3.2. Microbiota Composition Is Host-Specific and Varies between Diets

While α-diversity describes the general number of different taxonomies present in a sample and considers the evenness of their respective abundance, taxonomic profiling enables the visualization of the exact nature of these differences. Analysis on the genus level showed variations in the microbiota composition across diets (Figure 3A and Appendix A). In comparison with OV and PH, individuals who followed a VV diet harboured double to triple the relative amount of *Bifidobacterium* spp., *Prevotella* spp., and *Gemmiger* spp. within their intestine immediately after inclusion. *Prevotella* spp. could be detected in the OV group with a relative abundance of only 1.3%.

The mean relative abundance on the species level showed that the 12.1% of *Bifidobacterium* spp. in the VV consisted of 8% *Bifidobacterium adolescentis* (Figure 3B and Appendix A). After following the PH diet for at least four weeks, we detected a two-fold increase in *Bifidobacterium adolescentis* and *Coprococcus eutactus*. These changes were not identified as significant during differential abundance analysis. We further investigated the relative abundance for each individual on the PH diet at the time of inclusion in comparison with twelve weeks after (Figure 3C and Appendix A). Large variations in microbial composition between individuals at the time of inclusion could be observed, suggesting a partly host-specific microbiota composition.

We further analysed the differential abundance between OV and VV to highlight potentially interesting species, thereby only focusing on the top ten percent effect sizes (Figure 3D and Appendix A). We detected a 3-fold increase in *Prevotella copri*, a 4-fold increase in *Paraprevotella xylaniphila*, and an 18-fold increase in *Bacteroides clarus,* whereas, e.g., *Firmicutes bacterium CAG 94* showed a 6-fold decrease in the PH diet over the course of the study. The differential abundance depicted in Figure 3D and Appendix A suggests that following the PH diet shifts parts of the microbiota composition towards a VV microbiome. However, these observed changes were not significant.

Cultivation and species identification with MALDI-TOF mass spectrometry identified 59 different bacterial species across all time points among the five randomly selected participants from each group (Figure 4). Most commonly isolated were *Escherichia coli*, *Enterococcus faecium*, *Clostridium perfringens*, and *Bifidobacterium longum*. *Enterococcus mundtii* and *Priestia megaterium* were mostly detected in the VV, whereas *Streptococcus parasanguinis*, *Streptococcus salivarius*, *Enterobacter cloacae*, and *Bacteroides uniformis* were mainly isolated from faecal samples of those participants following the PH diet. A detailed account of detected bacteria in the different groups is displayed in Appendix A. This method, however, represents only cultivable microorganisms, leaving approximately 35–65% undetected when compared with next-generation sequencing (NGS) [35].

## 4. Discussion

Our study analysed whole-genome data obtained from faecal samples after following three specific diets, i.e., OV, VV, and PH, over the course of 12 weeks to investigate the intestinal microbiota composition associated with these dietary patterns. The main difference between OV as compared with VV and PH is most likely the intake of dietary fibre. Western citizens generally ingest between 14 g (United Kingdom) and 26 g (Norway) of dietary fibre, whereas most countries recommend 25–35 g per day for adults [36]. With the PH diet suggesting 232 g of whole grains, 300 g vegetables, and 200 g fruits per day for an intake of 2500 kcal/day, participants following this diet should reach these dietary fibre recommendations [2]. A sufficient amount of fibre is directly associated with positively affecting the human intestinal microbiome, and a plant-based diet is proposed to benefit human and planetary health [15,37]. In this study, we were able to detect a trend towards an increase in *Bifidobacterium adolescentis* and *Coprococcus eutactus* (Figure 3A–C) after following the PH diet for a minimum of four weeks. An increase in *B. adolescentis* has previously been shown after supplementation with inulin, a type of dietary fibre and naturally occurring plant carbohydrate. *B. adolescentis* is capable of degrading inulin into lactate and acetate, which can be used by *Anaerostipes hadrus* and *Enterococcus rectale* to produce the SCFA butyrate [38]. In contrast to the supplementation with inulin from Baxter et al., we did not find a co-increase in *A. hadrus* when following the PH diet without tracking the exact dietary fibre composition. However, *B. adolescentis* seems to have a growth advantage after increasing inulin intake. Similarly, β-glucans have been shown to be the preferred growth substrate of *C. eutactus*, suggesting a growth advantage after increasing β-glucans consumption [39,40]. Differences in taxonomic abundances suggested that several species merit particular consideration, such as, e.g., *Prevotella copri* and *Paraprevotella xylaniphila*, for which a non-significant increase was detectable (Figure 3D and Appendix A). *P. copri* is capable of dietary fibre degradation, as they harbour vast genomic repertoires of carbohydrate active enzymes [41]. Similar to *B. adolescentis*, switching to the PH diet might favour the growth of *P. copri.* While SCFA-producing bacteria should be beneficial for the host due to their anti-inflammatory and regulatory effects, *P. copri* has also been correlated with the development of rheumatoid arthritis, although without conclusive evidence. An overgrowth of *P. copri* might also inhibit the growth of other beneficial microbiota [42]. *P. xylaniphila* can produce anti-inflammatory SCFAs, but also has the potential to synthesise pro-inflammatory metabolites, such as, for example, succinic acid. Succinic acid was previously described in close correlation with the development of hypertension, inflammatory, and metabolic diseases [43,44]. These two species, identified by differential abundance, might harbour beneficial potential, but need to be studied more extensively to analyse their exact effect on health homeostasis and their function within the complex gut microbiome. However, computing the differential abundance is a powerful tool to identify both pathogenic species and beneficial bacteria. To the best of our knowledge, no genomic or phenotypic analyses have been performed to identify the biochemical properties of *Firmicutes bacterium CAG 94*, making this species an interesting target for further research.

Several limitations of our study are offered for consideration. First, we performed a monocentric analysis with a limited number of individuals. Second, participants of this study received recipes and detailed instructions on what to consume, but we did not implement exact meal plans. For future studies, we recommend standardised meal plans to avoid any potential participant compliance issues. Third, we did not perform culture-based bacteriological analysis in all study participants. Fourth, the VV contained mostly biologically female participants, thereby creating significant differences in sex ratios between the groups. Fifth, the study groups were relatively small, and robust statistical analyses of individual groups at different time points would require a larger study population in future studies.

In conclusion, this work provides the first metagenomics-sequencing-based appraisal of the PH diet. While no significant changes were observed within the overall intestinal microbial composition of individuals opting to follow the PH diet, we identified several potentially interesting bacterial species. Indeed, when focusing on differentially abundant species between OV and VV, non-significant trends of the PH cohort towards VV were noted. Specific bacterial species are capable of producing anti-inflammatory metabolites and might be an interesting target for novel probiotics, beneficial bacteria that can be taken supplementary to a healthy diet [45]. Hence, we encourage further microbiota-targeted research pertaining to the PH diet, ideally through multi-country longitudinal and larger-scaled studies.

## Figures and Tables

**Figure 1 nutrients-15-01924-f001:**
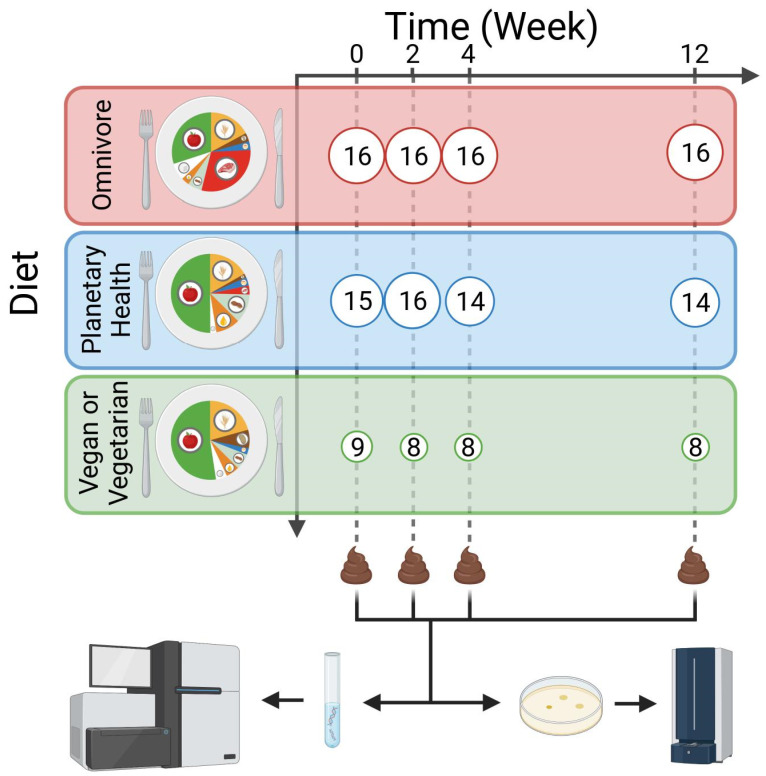
Design of the study. Participants followed three different diets over the course of twelve weeks. Stool was sampled at different time points and whole metagenome sequencing was performed. Additionally, bacteria were cultivated on different agar plates and analysed with MALDI-TOF mass spectrometry. Numbers in white circles depict the numbers of participants and respective stool samples at the different time points for each group.

**Figure 2 nutrients-15-01924-f002:**
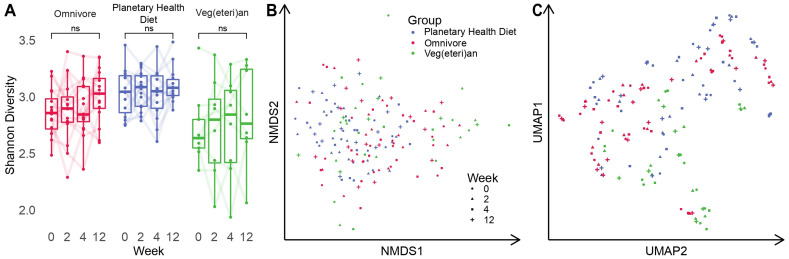
(**A**) α-diversity computed with the Shannon index for all time points and cohorts. Differences between initial and final time points were not significant for any cohort. ns = not significant (**B**) Visualised β-diversity computed with NMDS on weighted UniFrac distances among all sample pairs. (**C**) UMAP computed on sourmash distances computed among all samples.

**Figure 3 nutrients-15-01924-f003:**
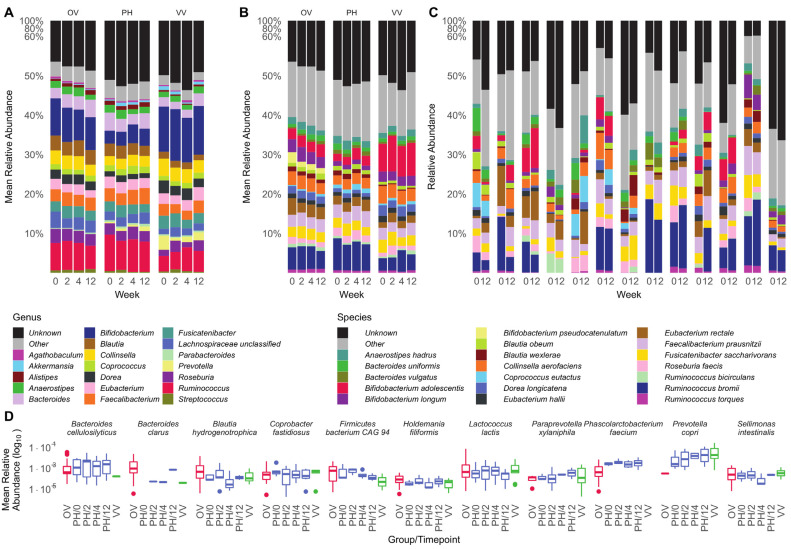
(**A**) Mean genus composition of the different dietary cohorts across different time points. Explicitly named genera were selected by looking at the highest mean relative abundances across all samples. (**B**) Identical information to panel Figure 3A, yet at species resolution. (**C**) Species composition of the PH cohort for the first and last measured time point. (**D**) Mean relative abundances of species with consistently largest effect sizes differentiating OV from VV. The results for the OV and VV cohorts were aggregated over all time points. OV, omnivore, VV, veg(etari)an, PH, Planetary Health group.

**Figure 4 nutrients-15-01924-f004:**
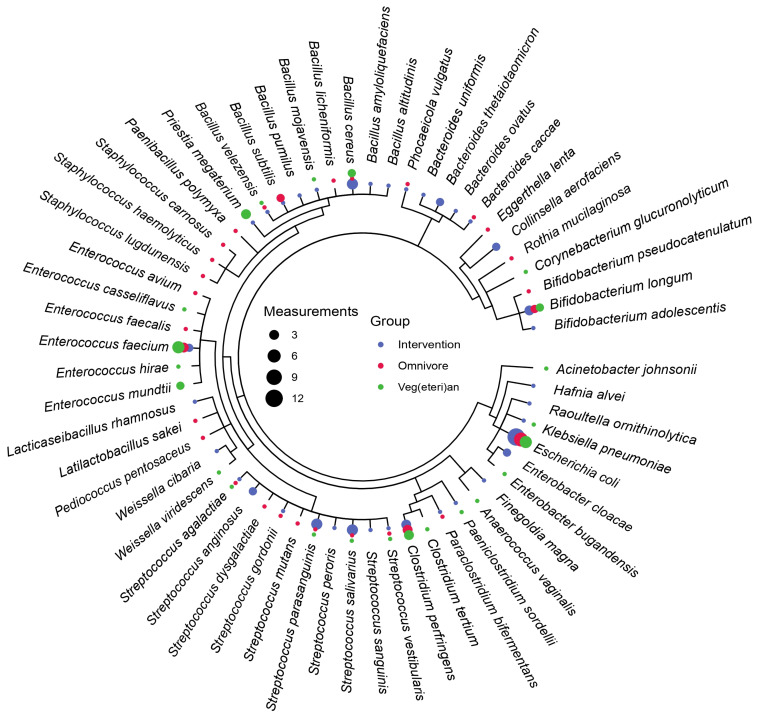
NCBI taxonomic classification of the species identified by mass spectrometry. The indicated number of measurements for the different diets represents the number of times the species has been identified in different samples at any time point.

**Table 1 nutrients-15-01924-t001:** General participant information. Listed below are the age ranges, BMI ranges, and sex ratio for all groups.

	OV	VV	PH
**Age ranges**	27–56	22–55	19–57
**BMI ranges**	19.8–32.8	19.9–40.1	20.0–24.4
**Male**	10	1	4
**Female**	6	8	12

## Data Availability

Deidentified participant data and study documents (clinical questionnaire, food documentation, informed consent form, and sequencing data) will be made available to other researchers upon reasonable request directed to the corresponding author via e-mail.

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
