# Peer review of "The Effect of a Planetary Health Diet on the Human Gut Microbiome: A Descriptive Analysis"

_nutrients, 2023, doi:10.3390/nu15081924_

Round 1

Reviewer 1 Report

Rehner et al., present a small (n = 41) longitudinal observation of healthy volunteers adhering to Planetary Health diet (PH – based on the EAT-Lancet), an omnivorous diet (OV) or vegetarian/vegan (VV). The authors observe changes across the 12 week study period, with increases in probiotic-associated bacteria in association with the PH diet, although few changes are significant. 

General comments:

The study has many strengths. Given the recent IPCC report, a shift to PH diets, in addition to other measures, will be necessary and hopefully encouraged in the coming years. Therefore it is a timely and important consideration to ensure that the necessary shift does not represent any unexpected health challenges. This study offers a small step in investigating this via the gut microbiome (a key biomarker of health). The use of shotgun metagenomics is welcome. The figures were appropriate and well designed. The study had appropriate exclusion criteria and considered control variables well, although there may have been an omission of BMI. Informed written consent was obtained from participants and the authors provide a REC reference for the study.

The study has some weaknesses. As the authors note, it is small in size and likely as a result there are few significant observations (the majority of trends discussed are not significant). There are additional details that are required in the methods to ensure the reproducibility of the study, and clarify the study design. A key concern is the inclusion in the abstract that ‘In the PH group, we observed a constant increase from 3.79 % at inclusion to 4.9 % after 12  weeks in relative abundance of Bifidobacterium adolescentis’ yet this change was not significant and this needs to be clearer.

Suggestions/Questions for the authors:

 The introduction provides a comprehensive introduction to the benefits of dietary fibre, but I felt that: a. The proposed hypothesis of a health benefit mediated by fibre and the gut microbiota due to an increase in fibre could have been clearer (if this was the authors intention), and b. that the divergence of the PH from the Mediterranean diet, to which it has several similarities and which has an established association to the GM could have been clearer. The authors talk about low-fat diets but it is not clear from this introduction why, unless they are relating this to the low levels of saturated fat in the Med/PH diets?  

How did the authors select which individuals participated in the intervention arm, and why did they decide against a paired (ie case-control) approach? Please add this detail to the study design section.

Did the authors measure and subsequently compare BMI/a measure of obesity between arms (as this has been shown to associate with the gut microbiome)?  

How did the authors ask participants to record their food in the two days prior to fecal sample (i.e. was a dietary software used?) and how was this analysed to ensure adherence? Please add this to the methods section.

Did the authors collect participant data on stool consistency which could be an important confounding factor (particularly given the increased fibre change in the intervention arm)?

Why did the authors decide to combine a metagenomics approach and bacterial culturing, given as the authors themselves note many bacteria are not culturable? Could the authors please clarify if the 5 samples they selected per diet group were aliquots rather than the full samples (ie all samples were analysed via who MGS?) If aliquots were used, were samples homogenised prior to aliquoting?

The data analysis section does not contain information on the statistical tests used to compare microbiome metrics of diversity between groups – please include this to ensure reproducibility.

Did the authors consider how library size might effect the diversity measures? Why did the authors not compare PH diets in their differential abundance analysis? Did the authors statistically compare group and time differences in beta diversity measures (eg via PERMANOVA etc?)

A table with descriptive statistics reflecting the first paragraph of the results would be welcome (ie age ranges per group, sex, BMI).

Were any side effects reported by participants following the dietary intervention? How many of the participants adhered to their diets?

Did the authors undertake a power calculation in designing their study to assess whether they recruited enough individuals to observe significant chances in the microbiota? If not, why was this number selected?

Would the authors in future consider including functional analysis in their work?

In the data availability statement, please may the authors include details of how researchers can make a data request? 

 Minor/Grammatical comments:

L43: The definition of ‘microbiome’ diverges from Marchesi’s: would the authors consider including the ‘collective genome’ in their definition: https://doi.org/10.1186/s40168-015-0094-5

L45-46 This sentence is hard to follow, would the authors consider breaking it up into two sentences?

L49 Additional comma required after for example, (would suggest rewording to ‘microbiome-influencing factors include the mode…’)

L50  - Would the authors consider rephrasing ‘female birth canal’ to ‘birth canal’ for inclusivity?

L75 - Suggest removing ‘for example’ for sentence flow

L77 – The authors suggest that low-fat diets are a ‘powerful method in weight management’ – is there evidence that this is true beyond short-term weight loss?

L117 – Would the authors consider rephrasing alcohol ‘uptake’ to ‘intake’

L172 – I believe the word ‘cores’ is included erroneously

L229 – α-diversity does not just describe the difference in ‘taxonomies’ present in a sample. The metric used here (Shannon diversity) considers both number of species and the evenness of their abundance.  

L251 – Would the authors consider rephrasing this to ‘these observed changes were not significant’

L289 – Would the authors consider rephrasing ‘a member of dietary fibre’ to ‘a type of dietary fibre’

The phrase ‘During differential abundance analysis’ is used a few times – would the authors consider rephrasing this to ‘Differences in taxonomic abundance suggested/suggests…

Author Response

Dear Reviewer,
we thank you very much for taking the time to help improve our manuscript. We uploaded the point-by-point response.

Reviewer 1

Rehner et al., present a small (n = 41) longitudinal observation of healthy volunteers adhering to Planetary Health diet (PH – based on the EAT-Lancet), an omnivorous diet (OV) or vegetarian/vegan (VV). The authors observe changes across the 12 week study period, with increases in probiotic-associated bacteria in association with the PH diet, although few changes are significant.

General comments:

→The study has many strengths. Given the recent IPCC report, a shift to PH diets, in addition to other measures, will be necessary and hopefully encouraged in the coming years. Therefore it is a timely and important consideration to ensure that the necessary shift does not represent any unexpected health challenges. This study offers a small step in investigating this via the gut microbiome (a key biomarker of health). The use of shotgun metagenomics is welcome. The figures were appropriate and well designed. The study had appropriate exclusion criteria and considered control variables well, although there may have been an omission of BMI. Informed written consent was obtained from participants and the authors provide a REC reference for the study.

The study has some weaknesses. As the authors note, it is small in size and likely as a result there are few significant observations (the majority of trends discussed are not significant). There are additional details that are required in the methods to ensure the reproducibility of the study, and clarify the study design. A key concern is the inclusion in the abstract that ‘In the PH group, we observed a constant increase from 3.79 % at inclusion to 4.9 % after 12  weeks in relative abundance of Bifidobacterium adolescentis’ yet this change was not significant and this needs to be clearer.

# We thank the reviewer for taking the time to read our manuscript and for making suggestions on how to add more value to our work. Especially, we thank the reviewer for pointing out the strengths and weaknesses of our study. We also believe, that the concept of the Planetary Health Diet is a valuable diet alternative in regard to global and individual health. We are happy to consider your following comments to improve the quality of our manuscript.

Suggestions/Questions for the authors:

 →The introduction provides a comprehensive introduction to the benefits of dietary fibre, but I felt that: a. The proposed hypothesis of a health benefit mediated by fibre and the gut microbiota due to an increase in fibre could have been clearer (if this was the authors intention), and b. that the divergence of the PH from the Mediterranean diet, to which it has several similarities and which has an established association to the GM could have been clearer. The authors talk about low-fat diets but it is not clear from this introduction why, unless they are relating this to the low levels of saturated fat in the Med/PH diets? 

# We appreciate the reviewer’s comments and questions about the introduction of the manuscript. To answer the first question, we believe that the main benefit of dietary fiber through the intestinal microbiome comes from its fermentation processes and the resulting production of short-chain fatty acids (SCFAs). We believe, that we elucidated this effect in line 51 to line 63 of the manuscript. To briefly summarize this paragraph: dietary fiber is fermented by several gut microbiota, which in turns produce SCFAs. These SCFAs have anti-inflammatory and anti-proliferative properties and are absorbed in the gut, reaching even distinct body sites, such as the lung. We used this example to show, that the positive effect of SCFAs does not stay locally in the gut, but can protect the respiratory tract against allergic reactions. We added a conclusive sentence in line 64 to line 66 in the manuscript in order to clarify the association with the Planetary Health Diet. We hope that this change is satisfactory to the reviewer.

To answer the reviewer's second comment, the PH and Mediterranean diet are indeed very similar. The Mediterranean diet allows more animal derived foods than the PH, however, the remaining food constellations are very much alike and mainly based on an increase in vegetables, legumes, whole-grain cereals, and fruit. As this proportion makes up almost half of each diet plan and is the major difference compared to the western diet, we used the Mediterranean diet, which is well studied, to elucidate the effect on an increase in dietary fiber intake that could be hypothesized for a very similar diet concept, the PH. We added this information of the major difference in line 71 to 73 of our manuscript and hope, that it sufficiently meets the reviewer’s expectations. In the introduction part, we wanted to point out the most commonly followed diet concepts, and those most frequently considered for global and individual health benefits. Therefore, we also included the low-fat diet. We are aware, that this diet concept differs significantly from the PH and Mediterranean diet. However, we believe that this information adds value to our introduction and would like to leave it included.

→How did the authors select which individuals participated in the intervention arm, and why did they decide against a paired (ie case-control) approach? Please add this detail to the study design section.

# Thank you for this question. We were dependent on the willingness of participants to change their diets according to the Planetary Health Diet, therefore, we did a wide survey and call on who would like to participate. We followed several exclusion criteria, for example the use of antibiotics within the last six months, pregnancy, and smoking to minimize potential bias. These are included in the study design section line 118-119. Considering the paired approach method, two concepts may be explored. On the one hand, participants may be paired up and each pair may be split into two different groups. Since literature describes differences in microbiome compositions between individuals to be quite considerable, we believe that this approach would incorrectly represent one of the major confounding factors. On the other hand, a pair may be defined as one individual at two different timepoints.  Most logically, this would correspond to the beginning and end of the study. Indeed, we present such an analysis in Figure 3C for all individuals of the PH as we hypothesized to see variations in this group after the course of 12 weeks. Further statistical evaluation remained unpaired due to sample mismatches that can be seen in Fig1. Further, we decided against this paired approach for the two control groups individually, as we expected to see minor variations due to other external factors, and did not see the need to depict these observations in detail for each individual as they are not the central objective of the study. Therefore, we chose to accumulate the data for each time point and each study group to show if major observations could be observed (Figure 3A on the genus level, Figure 3B on the species level). Since we did not observe major variations in microbial composition for these groups, we decided not to show the relative abundance for each individual as the focus is on the PH diet concept. We hope that this answers the reviewer’s question to their satisfaction.

We are unsure, on which detail the reviewer would like us to include to the study design section, as it describes how we designed the study and does usually not include why we decided against certain analysis approaches.

→Did the authors measure and subsequently compare BMI/a measure of obesity between arms (as this has been shown to associate with the gut microbiome)? 

# We thank the reviewer for their question. We collected patient information of each participant, including weight and height. To show variations between participants individually, we included Figure 3C, which shows the individual relative abundances of microbiota on the species level for each participant of the PHD group. We also calculated the BMI for each participant. The average BMI value was 28.8 and ranged from 19.8 to 40.1. However, most individuals included (n=31) had a normal BMI. Eight individuals had a BMI between 25.1 and 33 (overweight, obese class I), and one individual had a BMI of 40.1 (obese class III). The individual showing a BMI of 40.1 was part of the vegan/vegetarian group. This group was very limited in numbers, as we were unable to identify many individuals following either of these two diet concepts who were willing to participate, we still included this individual in the analysis. We are aware, that either obesity can alter the microbial composition or the microbial composition can contribute to obesity. However, we believe that this effect is marginal as the results were accumulated from all participants of this group. Furthermore, the selected participants represent a random sample of western world citizens, and the main focus of the manuscript lies on the effect of the PH diet concept.

→How did the authors ask participants to record their food in the two days prior to fecal sample (i.e. was a dietary software used?) and how was this analysed to ensure adherence? Please add this to the methods section.

# We thank the reviewer for his valuable suggestion. We added this part to the methods section line 136 to line 138.

Briefly, we printed food diaries for each participant, asking them to track every divergence they may have had from the PH diet. From the control groups and the PH, we asked for a detailed description of food intake two days prior to stool sampling.

→Did the authors collect participant data on stool consistency which could be an important confounding factor (particularly given the increased fibre change in the intervention arm)?

# We appreciate the reviewer’s suggestion. We did not collect data on stool consistency, however we will track this factor for all following studies and thank the reviewer for their excellent comment.

→Why did the authors decide to combine a metagenomics approach and bacterial culturing, given as the authors themselves note many bacteria are not culturable? Could the authors please clarify if the 5 samples they selected per diet group were aliquots rather than the full samples (ie all samples were analysed via who MGS?) If aliquots were used, were samples homogenised prior to aliquoting?

# This is indeed a very interesting question, that we will gladly answer. During our first microbiome studies, we used the culturing of bacteria as a quality control of the sample, as well as a quality for the used database for taxonomic profiling of the sequencing data. We noticed that some bacteria were identified by culturing and subsequent MALDI-TOF analysis but were missing in the sequencing data after taxonomic profiling. We identified missing bacterial species in the database we used and decided to keep performing both experiments in order to avoid false representations of the microbiome and allow for quality control of the sample and used reference databases.

To answer the second part of your question, we analysed all samples via sequencing and randomly selected five samples per group as representatives for the quality of the sample and to control our computational analysis. All stool samples were vortexed after defrosting in order to homogenize the sample without mechanically lysing bacterial cells. Of these homogenized samples, we used 50 mg for DNA extraction, and 5 mg for culturing. We added this information to our manuscript in line 168 to 170.

→The data analysis section does not contain information on the statistical tests used to compare microbiome metrics of diversity between groups – please include this to ensure reproducibility.

# We thank the reviewer for their comment and would like to point out, that for α-diversity, we performed the Wilcoxon rank sum test on Shannon diversity, for which we did not observe significant p-values. This is mentioned in the figure legend 2A. We moved this information to the methods section for clarity. Considering beta-diversity, see the next response.

→Did the authors consider how library size might effect the diversity measures?

# In one of our pilot studies, we explored the impact of library size by in silico downsampling the data. Based on our findings of this first study, we concluded the settled to aim for a sequencing depth of 5GB for studies moving forward (doi: 10.1016/j.gpb.2022.05.006.). Considering statistical analysis rarefication to equal library size is not recommended in literature (.https://doi.org/10.1371/journal.pcbi.1003531).

→Why did the authors not compare PH diets in their differential abundance analysis?

# We thank the reviewer for their question and would like to point out, that with only 16 participants in the PH we could only perform differential abundance with 15 samples at inclusion versus 14 samples after 12 weeks of the study. With this sample set, the lack of statistical power is unlikely to reveal statistically significant changes in species during differential abundance analysis. In order to overcome this shortcoming of the study, we decided to use all samples of all time points from the OV and VV and thereby detect species that differ between the two groups. Next, we wanted to see if following the PH for 12 weeks leads to changes in those highlighted species.
In a next study design, in which we would like to enlarge the participant groups considerably, we will of course also compare differentially abundant species within the PH group. We highlighted the small study groups as considerable limitations in our revised ‘Discussion’ (see revised manuscript, lines 354-356).

 →Did the authors statistically compare group and time differences in beta diversity measures (eg via PERMANOVA etc?)

# PERMANOVA analysis was performed on weighted Unifrac distances at one thousand permutations. With individuals appearing multiple times in the same group at different timepoints, p-values will be heavily inflated. Subsetting by timepoint to avoid this issue only leads to one significant p-value = 0.014 at study start. This result becomes insignificant after adjustment for testing each week. Comparison within each cohort comparing initial versus final timepoint suffers of low sample size and no significant results were detected.

→A table with descriptive statistics reflecting the first paragraph of the results would be welcome (ie age ranges per group, sex, BMI).

# We thank the reviewer for his great idea to add a table with general participant information. We agree, that this would indeed add value to the manuscript and therefore added such a table (Table 1) in line 236 to the first paragraph of the results section.

→Were any side effects reported by participants following the dietary intervention? How many of the participants adhered to their diets?

# The participants did not report any side effects following the Planetary Health Diet. However, this can differ individually. If an individual switches to a high fiber diet radically, and increase dietary fiber intake by 20 g from one day to another, side effects such as bloating, constipation, or diarrhea may occur. However, none of the participants included in this study reported such conditions. To answer the second part of the question, all 16 participants of the intervention group adhered to their diets without exceptions during the course of 12 weeks. We would like to note, that if a participant would have wanted to be excluded and stop the PH diet concept, they could have notified us and stop the study. This part was included in the informed consent form. However, all individuals participated for the entire 12 weeks and followed the recommendations by the EAT-Lancet commission.

→Did the authors undertake a power calculation in designing their study to assess whether they recruited enough individuals to observe significant chances in the microbiota? If not, why was this number selected?

# We thank the reviewer for this excellent question. Indeed, for this proof-of-concept study, we did not perform a power calculation to achieve significant results. This study did not yield significant changes, however implicate trends after following the PH for 12 weeks. This trend is visible with the selected amount of participants. Studies including humans always depend on the willingness of individuals to participate. We further believe, that the microbiome is as individual as the human fingerprint, therefore changes can also be very individual, which is why we chose to include Figure 3C. We would like to bring the idea to the scientific community, that the PH concept, as it proposes to be beneficial for global health and individual health, does have an impact on the human intestinal microbiome and provide first information on how this impact may look like. As a proof-of-concept study, we are planning on starting a new study with improved food tracking, stool consistency tracking, and larger group sizes that will be set by power calculation in designing the next study. This study was a monocentric study, which we plan to enlarge to at least further regions in Germany, but also different countries. In order to reduce the effect of the relatively small sample size, we used very strict exclusion criteria. We hope that this answers the reviewer’s question to their expectations.

→Would the authors in future consider including functional analysis in their work?

# Functional analysis indeed is a very interesting aspect to consider during microbiome studies. We started a very close collaboration with the Helmholtz Institute for Pharmaceutical Research Saarbrücken (HIPS) to, for one more closely investigate biosynthetic gene clusters, which often translate to interesting metabolites, as well as next to an existing antimicrobial resistance pipeline, an antimicrobial resistance prediction pipeline. As we perform metagenomic sequencing posterior to whole-genome DNA extraction, we would like to use the entire information we can yield from the sequencing data for future studies, including functional analysis to better study the biochemical potential and capacity the detected microorganisms have. 

→In the data availability statement, please may the authors include details of how researchers can make a data request? 

# We apologize that the phrasing was not completely clear. We added this information in line 386 of the Data Availability Statement. Researchers can contact the corresponding author (Prof. Dr. Dr. Sören Becker) via the e-mail address in the corresponding author section and ask for the data.

 Minor/Grammatical comments:

→L43: The definition of ‘microbiome’ diverges from Marchesi’s: would the authors consider including the ‘collective genome’ in their definition: https://doi.org/10.1186/s40168-015-0094-5

# Thank you for pointing out the missing information about the microbial genome. We added this information in line 44 and added the suggested reference to the manuscript as reference number 3.

→L45-46 This sentence is hard to follow, would the authors consider breaking it up into two sentences?

# We agree that this sentence was difficult to follow and divided it into two sentences. The reviewer can find the change in line 45.

→L49 Additional comma required after for example, (would suggest rewording to ‘microbiome-influencing factors include the mode…’)

# We appreciate the suggestion and gladly follow the reviewer’s advice. We changed the sentence according to their phrasing in line 50.

→L50  - Would the authors consider rephrasing ‘female birth canal’ to ‘birth canal’ for inclusivity?

# We agree there was no need to specify on this information and removed “female” in this sentence.

→L75 - Suggest removing ‘for example’ for sentence flow

# The suggestion of removing the phrase “for example” indeed improves the sentence flow. Therefore, we decided to follow the reviewers suggestion in line 82.

→L77 – The authors suggest that low-fat diets are a ‘powerful method in weight management’ – is there evidence that this is true beyond short-term weight loss?

# That is indeed a very good question. To our knowledge, it is debatable how well diet concepts work in terms of weight loss. We believe, that every rather balanced diet concept has the potential to aid in weight reduction, however, strongly dependent on the intensity of the diet concept and the discipline of the individual. Tobias et al. Very nicely pointe out, that a low-fat diet showed no superior effect in terms of weight loss compared to other diet concepts (doi: 10.1016/S2213-8587(15)00367-8). A more recent study from 2022 described several positive effects of a low-fat diet in individuals following this diet for a minimum of 6 months (https://doi.org/10.3389/fnut.2022.935234), however there is still need of actual long-term effect studies.

→L117 – Would the authors consider rephrasing alcohol ‘uptake’ to ‘intake’

# We changed the wording to intake, as we also believe that this change makes sense.

→L172 – I believe the word ‘cores’ is included erroneously

# We apologize for not noticing the falsely incorporated word. The reviewer correctly identified the word as erroneously included. We appreciate the detailed analysis of our manuscript and removed the word “cores” in now line 193.

→L229 – α-diversity does not just describe the difference in ‘taxonomies’ present in a sample. The metric used here (Shannon diversity) considers both number of species and the evenness of their abundance. 

# Thank you for your valuable comment. We agree, that Shannon Index does not only compute the differences in which species are present, but as the reviewer suggest also includes the evenness of their abundance. We appreciate the suggestion and added this information to the description of the α-diversity in line 260.

→L251 – Would the authors consider rephrasing this to ‘these observed changes were not significant’

3 We did consider rephrasing the sentence and changed it according to the reviewer’s suggestion. You can now find the sentence in line 282.

L289 – Would the authors consider rephrasing ‘a member of dietary fibre’ to ‘a type of dietary fibre’

# We agree that “a type” is the better expression than “a member” and changed the wording according to the reviewer’s comment in line 319.

→The phrase ‘During differential abundance analysis’ is used a few times – would the authors consider rephrasing this to ‘Differences in taxonomic abundance suggested/suggests…

# We thank the reviewer for their suggestion on rephrasing this particular phrase and changed it in line 327. We hope that this change is satisfactory to the reviewer, as we only use this exact wording now twice in the manuscript.

Reviewer 2 Report

In the manuscript, Rehner and colleagues investigate intestinal microbiota of participants on either an omnivore (OV), vegetarian/vegan (VV) or Planetary Health (PH) diet. The manuscript is clearly and concisely written, the results are adequately presented and, from the perspective of intestinal microbiota, support PH diet as a viable alternative to other diets based around a healthy lifestyle.

-        - in the introduction, Mediterranean Diet is described. As this diet is not the focus of the current study, it may confuse the reader. While this diet can be mentioned as an alternative to either VV, PH or other diets, the focus should stay on PH. Apart from that, the differences among OV, VV and PH should be specified more clearly. The first paragraph could be divided into more paragraphs to facilitate easier readability (for example, the information regarding SCFAs could be in a separate paragraph)

-       - line 47: If authors write “several recent studies” in the text, then it is expected that they cite several studies instead of just one, I recommend the authors to either add more citations or change the text accordingly

-       - line 57: “SCFAs have been shown to influence glucose…” this claim contains no citation

-      - what do the numbers in circles in Figure 1 mean, the number of participants/samples for each collection/time point? If so, I would advise to add a brief explanation in the figure legend

-       - I would advise the authors to keep group naming consistent, e.g. to name the experimental group as the “PH” group; this can be seen in some figures such as Figure 3A, 3B but in Figure 2 the group is called “intervention”

-       - although I appreciate that several limitations were stated in the discussion, the authors did not explain if there was a reason behind having a grouped vegetarian/vegan group; having participants with different dietary habits in the same group could have contributed to the observed variability within the group

-       - if similar studies will be done by the authors in the future, it would be interesting to see any differences in intestinal microbiota of an experimental group changing diet from VV to PH

-       - I would suggest the authors to check spelling in the manuscript, unite the usage of English (either British or American, not both); I found these typographical errors:

line 42: “digestibale

line 44: “microorganismns”

line 51: “focussing” (this error appears in multiple instances throughout the manuscript)

line 58: “feaces”

line 135: “resepctive”

Author Response

Dear Reviewer,

we thank you very much for taking the time to help improve our manuscript substantially. We uploaded the detailed point-by-point response to your comments and suggestions.

Comments and Suggestions for Authors

In the manuscript, Rehner and colleagues investigate intestinal microbiota of participants on either an omnivore (OV), vegetarian/vegan (VV) or Planetary Health (PH) diet. The manuscript is clearly and concisely written, the results are adequately presented and, from the perspective of intestinal microbiota, support PH diet as a viable alternative to other diets based around a healthy lifestyle.

# We appreciate the positive feedback of the reviewer and thank them for highlighting the potential importance of our manuscript to the scientific community. We further thank the reviewer for taking the time to read our manuscript in detail and make valuable suggestions and comments. We hope that we modified the manuscript according to the reviewer’s expectations.

-       - In the introduction, Mediterranean Diet is described. As this diet is not the focus of the current study, it may confuse the reader. While this diet can be mentioned as an alternative to either VV, PH or other diets, the focus should stay on PH. Apart from that, the differences among OV, VV and PH should be specified more clearly. The first paragraph could be divided into more paragraphs to facilitate easier readability (for example, the information regarding SCFAs could be in a separate paragraph)

# The reviewer made some excellent suggestions, which we are happy to follow in our manuscript. Therefore, we divided the introduction in topic-related sub-paragraphs, for example in line 42, line 55, line 67, and line 76. We hope these changes facilitate an easier readability for the introduction.
Furthermore, the reviewer pointed out correctly, that a lot of information is given about the Mediterranean Diet. This holds true, as is this one of the best studied diet concepts, which many clinicians recommend for a healthy lifestyle. Also in regard to the intestinal microbiome, many studies have investigated the effect of the Mediterranean Diet. Since this diet is very similar to the PH, we decided to keep the information in the manuscript, but also included the major difference between the Mediterranean Diet and the PH in line 71-73. We hope, that this meets the reviewer’s expectations and that our reasoning is acceptable. We further specified the OV and VV diet concepts in the introduction in line 103 to line 112.

-       - line 47: If authors write “several recent studies” in the text, then it is expected that they cite several studies instead of just one, I recommend the authors to either add more citations or change the text accordingly

# We thank the reviewer for pointing out the controversial statement we used in line 47-48. We changed the phrasing accordingly and hope this change is satisfactory to the reviewer.

-       - line 57: “SCFAs have been shown to influence glucose…” this claim contains no citation

# We apologize for not including the reference. We have now added the reference to the manuscript with the number 8, doi: 10.3390/ijms21176356.

-      - what do the numbers in circles in Figure 1 mean, the number of participants/samples for each collection/time point? If so, I would advise to add a brief explanation in the figure legend

# The suspected information behind the numbers in white circles in Figure 1 are correct. We thank the reviewer for pointing out that this explanation would add valuable information the manuscript and included it in the figure legend of Figure 1. The reviewer can find this information now in line 143-144.

-       - I would advise the authors to keep group naming consistent, e.g. to name the experimental group as the “PH” group; this can be seen in some figures such as Figure 3A, 3B but in Figure 2 the group is called “intervention”

# The reviewer’s comment is much appreciated. In Figure 3 we used the abbreviations for each study group, that were introduced in the methods section. In terms of available space and a visually attractive design, we chose to keep the abbreviations in this figure. In Figure 2, we changed “Intervention” to “Planetary Health Diet” for easier understanding and uniformity in group descriptions. We hope this change is satisfactory to the reviewer.

-       - although I appreciate that several limitations were stated in the discussion, the authors did not explain if there was a reason behind having a grouped vegetarian/vegan group; having participants with different dietary habits in the same group could have contributed to the observed variability within the group

# We thank the reviewer for pointing out this limitation and would like to explain the reason for grouping vegetarians and vegans: unfortunately, we reached very little individuals following these two diet concepts. The initial plan was to have four distinct groups, dividing vegans and vegetarians. However, we only found 9 individuals in total that followed either a vegan or vegetarian diet. Since these two diet forms are similar to each other in terms of meat consumption, and differ significantly from the omnivorous diet, we decided to group them together as the sample size was very low. For future studies, we hope to be able to include more participants for each group and divide vegans and vegetarians into two groups. Since we stayed locally with our study and did not expand to other regions or even countries, unfortunately we had to accept this limitation of our study. However, as a proof-of-concept study that has a focus on the planetary health diet, we believe that the limitation is acceptable and can only be improved in the future. Yet, we were able to observe a trend in microbial composition in the gut after following the PH for 12 weeks.
We appreciate the comment, however we do not discuss the variations observed during beta-diversity analysis and therefore did not include this grouping as a possible reason behind no clear cluster formation.

-       - if similar studies will be done by the authors in the future, it would be interesting to see any differences in intestinal microbiota of an experimental group changing diet from VV to PH

# We believe that this indeed would be very interesting to observe. Since animal-derived foods are allowed within the concept of the PH, it would have to be an adjusted version excluding the animal-derived foods from the PH, as it might be challenging and unethical to convince vegans and vegetarians to include meat and/or dairy products to their diet. However, many vegans and vegetarians tend to eat “unhealthy” diets in the western world, for which reason it would be very interesting to simply increase their dietary fibre intake and observe potential changes according to the suggested nutrient ratios by the EAT-Lancet commission, and decrease any sort of highly processed foods within their diet (e.g. vegan sweets).

-       - I would suggest the authors to check spelling in the manuscript, unite the usage of English (either British or American, not both); I found these typographical errors:

# We thank the reviewer for their suggestion on typographical errors. We checked for spelling in the manuscript with the word editor function and could not detect any more spelling mistakes. Furthermore, we checked for American and British English and should now have a uniform usage following British English.

line 42: “digestibale”

# Thank you for pointing out this mistake. We changed the word to its correct form “digestible” in line 42.

line 44: “microorganismns”

# We apologize for the inconvenience and not detecting this misspelling before submission. We changed the word accordingly in line 44.

line 51: “focussing” (this error appears in multiple instances throughout the manuscript)

# We thank the reviewer for detecting this spelling mistake and changed it in the three positions in the manuscript: line 52, line 274, and line 356.

line 58: “feaces”

# We are deeply sorry for not noticing this spelling mistake before submitting the manuscript to Nutrients and are grateful the reviewer took the time to, in detail, screen for spelling errors throughout the manuscript. We gladly changed the word in line 59.

line 135: “resepctive”

# We changed the word and thank the reviewer for pointing out the mistaken spelling in line 154.